# The ancient and helical architecture of *Elasmobranchii's* spermatozoa enables progressive motility in viscous environments

**Sergii Boryshpolets** [1]*, **Borys Dzyuba**[1], **Pablo García-Salinas**[2],
**Hermes Bloomfield-Gadêlha**[3], **Victor Gallego**[2], **Anatoliy Sotnikov**[1], **Juan F. Asturiano**[2]

**1** Faculty of Fisheries and Protection of Waters, University of South Bohemia in České Budějovice, South Bohemian Research Center of Aquaculture and Biodiversity of Hydrocenoses, Vodňany, Czech Republic, **2** Grupo de Acuicultura y Biodiversidad, Instituto de Ciencia y Tecnología Animal, Universitat Politècnica de València, Valencia, Spain, **3** School of Engineering Mathematics and Technology & Bristol Robotics Laboratory, University of Bristol, Bristol, United Kingdom

* sboryshpolets@frov.jcu.cz

## Abstract

Subclass *Elasmobranchii* belongs to an old evolutionary class of Chondrichthyes that diverged 450 mya, presenting a wide diversity of reproductive strategies while preserving the ancient mode of internal fertilization. Despite such evolutionary success, many species in this group are at serious risk of extinction. Understanding the principles of sperm progressive motility and physiology of such an ancient group of vertebrates is crucial for advancing future assisted reproductive techniques to safeguard this species and for deepening our understanding of the evolution of reproduction. *Elasmobranchii* species possess big spermatozoa (compared to bony fishes) with an elongated helical head and tail similar to one currently existing (but later diverged) in birds, reptiles, and amphibians, which can be considered an evolutionary ancient. These structures may be associated with the necessity to penetrate viscous ovarian fluid or the jelly layer of eggs, suggesting environmental viscosity as the driving pressure shaping large-sized sperm heads into helical shapes through evolution. We observed spermatozoa motility with high-speed video microscopy to capture sperm and flagellar motion in three *Elasmobranchii* species: the freshwater ray *Potamotrygon motoro*, the marine skate *Raja asterias* and the shark *Scyliorhinus canicula*. We investigated the effect of viscosity on spermatozoa motility parameters and its ability to break free from spermatozeugmata, move progressively, and perform directional changes. After 20 min of observation, the spermatozeugmata conserved their structure in a low viscosity media of 1000 mOsm/kg osmolality. In comparison, no remaining structure of spermatozeugmata could be found in high-viscosity media with 2% methylcellulose (MC) in all three species due to progressive spermatozoa motion. We find that spermatozoa's unique helical head-to-flagellum architecture is specific to promote locomotion in high-viscosity fluid; they cannot move progressively in low viscosity. The highest velocity for shark sperm was observed at 0.75% MC and 1% MC for ray and skate sperm. Viscosity stabilizes the flagellar propagation, producing rotational forces and allowing the helical head to "screw" into the media. Our observations suggest that the surrounding viscosity is critical to allowing spermatozoa progressive motility and enabling

**Data availability statement:** All relevant data are within the manuscript and its Supporting Information files.

**Funding:** Part of the work was carried out with the support of: VVI CENAKVA Research Infrastructure (ID 90099, MEYS CR, 2019–2023); "Biodiversity" (CZ.02.1.01./0.0/0.0/16_025/0007370 Reproductive and genetic procedures for preserving fish biodiversity and aquaculture), and the Project PID2022-138847-I00 funded by MICIU/AEI/10.13039/501100011033 and ERDF/EU. VG has a "Beatriz Galindo" aid (BG22/00024; for the attraction of research talent—2022) funded by the Ministerio de Universidades (Spain). The funders had no role in study design, data collection and analysis, decision to publish, or preparation of the manuscript.

**Competing interests:** The authors have declared that no competing interests exist.

spermatozoa to control direction via newly observed head buckling in high viscosity. As such, the viscosity may be a key element controlling and regulating sperm performance and navigation during fertilization in the *Elasmobranchii* species.

## Introduction

Chondrichthyans (cartilaginous fishes, including *Elasmobranchii*) are an old evolutionary class of aquatic vertebrates that diverged 450 mya [1]; Fig 1). Currently living cartilaginous fishes are characterized by various reproductive strategies [2] while still being unique since they preserved an ancient mode of internal fertilization in contrast to bony fishes, in which internal fertilization appeared evolutionary later, from externally fertilizing ancestors [3]. As such, the cartilaginous, particularly *Elasmobranchii* fish (most commonly known as sharks and rays), are critically important for reproductive biology studies, from its spermatology, physiology, and biophysics since their reproduction strategy remained almost unchanged in these species for millions of years.

Spermatozoa are one of our planet's most diverse eukaryotic cell types [4]. While highly specific for reproduction purposes (its primary function is to deliver the male genetic information to the female ova), spermatozoa are taxa–specific and have morphed as a response to different fertilization and environmental pressures. This is the fundamental hypothesis of Darwinian sex evolution. It is predicted that due to gamete competition, individuals who start

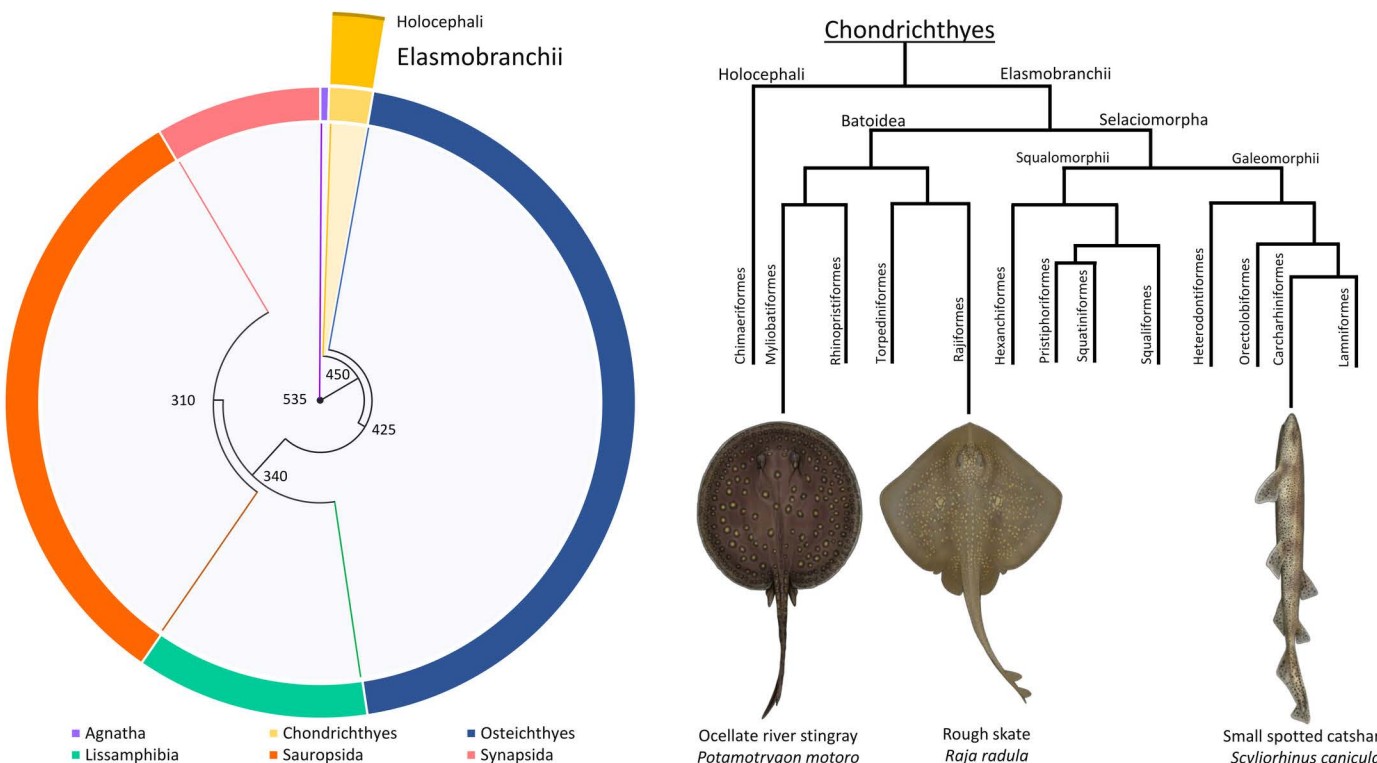

**Fig 1. Cladogram of the relationship between present-day vertebrates.** The length of the outer ring segments shows the number of extant species. The number next to the nodes refers to the last common ancestor in millions of years. The yellow portion represents the chondrichthyans, consisting of the *Elasmobranchii* and the *Holocephali*, whose proportional representation is based on their number of extant species. The phylogenetic tree is adapted from Amaral et al., 2018. The three species illustrated, *Potamotrygon motoro*, Raja radula, and *Scyliorhinus canicula,* are placed next to the order to which they belong.

to produce larger amounts of smaller-sized cells become defined as a male, which competes for fertilizing big-size ova produced by females, providing a safe environment for the nourishment of the future embryo and its development.

Several studies have demonstrated that different fertilization modes could drive the overall length of the spermatozoa: longer cells are correlated with internally fertilizing species and shorter cells with externally fertilizing ones [5]. As internal fertilization can be considered an evolutionary primitive feature for all gnathostomes [6], the evolutionary studies of early diverged taxa, such as cartilaginous fishes, are of primary interest for understanding the fundamental pathways leading to the diversity of reproduction modes in vertebrates and taxa specificity of spermatozoa morphology. Interestingly, some evolutionary later diverged birds [7], reptiles [8], monotremes [9–11], and amphibians [12] share a similar sperm morphology to that of cartilaginous fish, characterized by elongated and helical-shaped heads.

This similarity hints that chondrichthyan fishes' spermatozoa shape is evolutionarily ancient (plesiomorphic for vertebrates). Their distinctive features, such as sperm head morphology and flagellar structures, could be a specific adaptation to allow sperm to penetrate the highly viscous environment of the ovarian fluid and the jelly layer that coats the eggs. If this is the case, the environmental fluid viscosity could be the main driving force in shaping large-sized spermatozoa heads into slender helical-shaped heads during evolution.

Generally, hydrodynamical effects associated with body shape and size are well known to influence the swimming kinematics of microorganisms [13]. The sperm head shape was recently found essential for progressive motility and spinning movement in two cartilaginous fishes, rays- *Rhinoptera javanica* and *Taeniura meyeni*, possessing the head helical shape with four turns. This head shape was experimentally and computationally found essential for adaptation to a viscous environment. Contradictory statements have been made whereby the low progressive motion of sperm has been observed in high-viscosity conditions [14].

Despite valuable knowledge of the processes of fertilization and reproduction, there is still no clear understanding of how viscosity could influence sperm motility in different species, particularly in relation to the size and shape of spermatozoa. Thus, future studies of viscosity's biological relevance and its effect on reproduction are still highly interesting. Several studies have examined the impact of swimming media viscosities on sperm performance in different species, using different substances to alter the viscosity. Based on previous detailed studies [15,16], we choose methylcellulose due to its non-toxic, inert nature, stability and ability to modify the viscosity of solutions in relatively low concentrations.

Moreover, it is unclear whether the effect of viscosity is specific to all *Elasmobranchii* and other species with helical spermatozoa architecture, with a big diversity of spermatozoon shape and size [17]. Other studies suggest that sperm competition associated with postcopulatory sperm selection also affects the sperm flagellum length in sharks, though the reasons for diversity in spermatozoon head and midpiece shape and length remain unclear [18]. Understanding the spermatozoon structure and principles of sperm motility and physiology is an essential step toward better predicting the different pressures driving their evolutionary adaptations. Also, from a more pragmatic point of view, this knowledge may be indispensable for developing the next generation of assisted reproductive techniques in *Elasmobranchii*, as many of them face extinction [19].

The study aims to investigate the effect of surrounding fluid viscosity on spermatozoa motility parameters and its ability to swim progressively in connection to the specific sperm head shape and size. To achieve this, get a better understanding of the role of viscosity across different species, and gain more insights into the reproduction process in *Elasmobranchii*, we investigated the sperm motility and flagellar propagation in three species: (1) Small-spotted catshark - *Scyliorhinus canicula,* (2) Mediterranean starry skate – *Raja asterias* and (3) the South

American stingray - *Potamotrygon motoro* as a representative of freshwater later diverged *Elasmobranchii* species (Fig 1).

## Results

### Spermatozoa structure

The *Elasmobranchii* spermatozoa possesses a long and helical head, an elongated midpiece, and a flagellum supplemented with additional ultrastructural components to its axoneme. The head shape of the ray and skate spermatozoa observed are helixes turning toward the left (counterclockwise helixes, with a large non-monotonic modulated pitch and diameter along its length, increasing to a maximum before decreasing again at the connecting piece (Fig 2A). Skate spermatozoa had a head with a length of $48.16 \pm 1.31$ μm, helix amplitude of $4.48 \pm 0.35$ μm, and wavelength of $11.33 \pm 0.75$ μm possessed 4,5 helix and were similar in all 4 males. Ray spermatozoa had larger heads, with a length of $54.21 \pm 1.09$ μm, helix amplitude of $5.32 \pm 0.17$ μm, and wavelength of $10.63 \pm 0.6$ μm possessed 5,5 helix and also were similar in all 3 males. The shark (*Scyliorhinus canicula*) spermatozoa observed, on the other hand, is a "screw-shaped" helix turning toward the left (counterclockwise helixes) with approximately 30–35 turns, accurate measurement of which was challenging due to a tiny diameter ($3.28 \pm 0.66$ μm), pitch ($2.23 \pm 0.27$ μm) that remains constant along its length $59.36 \pm 1.51$ μm (Fig 2B).

In all species, the flagellar ultrastructure is reinforced by two accessory axonemal columns on the sides of the axoneme (Fig 2C). These columns rotate clockwise relative to the 9 + 2 axoneme,

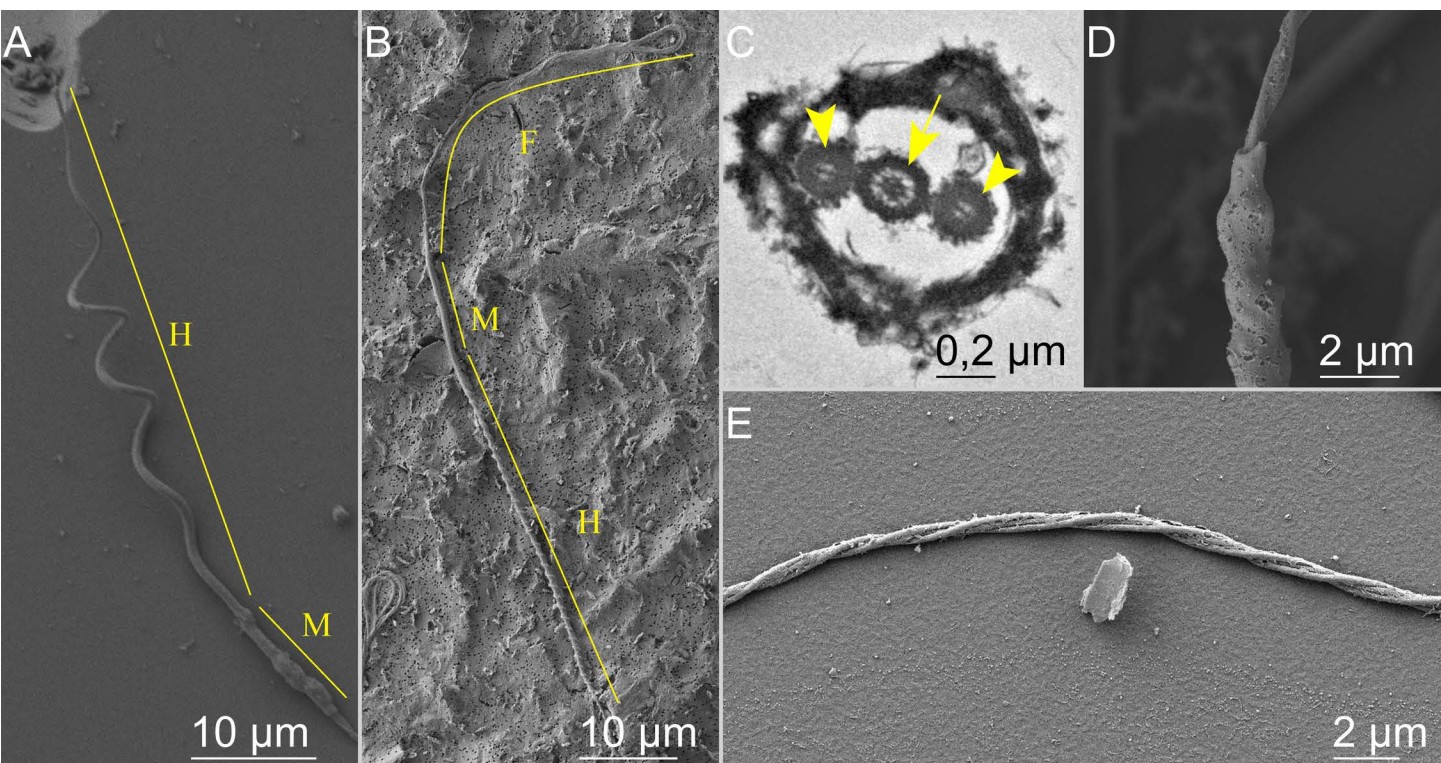

**Fig 2. Electron microscopy micrographs of spermatozoon structures.** (A) - ray *Potamotrygon motoro* spermatozoon head (H) and midpiece (M), (B) - shark *Scyliorhinus canicula*, SEM of the whole spermatozoon: head (H), midpiece (M) and flagellum (F). (C) – element of the flagellum in ray *Potamotrygon motoro*, transverse section of the flagellum, TEM, arrowheads - longitudinal columns, arrow – 9 + 2 structure axoneme. (D) - element of midpiece in ray *Potamotrygon motoro*, SEM. (E) - element of the flagellum in ray *Potamotrygon motoro*, SEM.

following the same handedness of their helical heads. Thus, they are twisted around the axoneme in a counterclockwise helical fashion (Fig 2E). A similar twisting is observed at the midpiece (Fig 2D). Interestingly, the counterclockwise helical chirality of head-to-flagellum structures is conserved for all three species despite the changes in head morphology between rays/skates and sharks.

## Spermatozeugmata: Collective beating sperm bundles in large sperm clusters

We observed spermatozeugmata for all three species, sperm not encapsulated and tails of peripheral sperm protruding in all cases (Fig 3). The spermatozoa arrangement within the spermatozeugmata is very distinct between ray and skate (Fig 3A, 3B). Spermatozeugmata in rays contain different amounts of randomly distributed spermatozoa, forming a mesh-like structure (Fig 3A). Skate sperm clusters ranged from dozens to hundreds of sperm cells (Fig 3B). Spermatozoa were all braided and coiled together by the helical heads, aligned in parallel so that all heads seemed to work as one unit. The parallel alignment permitted the self-organized formation of large flagellar bundles that could beat in collective synchrony with high levels of order, with travelling waves propagating in the bundle from head to tail. Shark spermatozoa were connected by the tip of the heads and at the midpiece region, forming a symmetric arrangement of their cluster towards a common centre. As such, sperm heads tended to be directed to the centre while their moving flagellum formed a large-scale beating flagellar bundle in the outer direction (Fig 3C).

## Sperm release from spermatozeugmata clusters

Spermatozeugmata in all three species were observed to possess active flagellum within the high-density clusters of spermatozoa. In a low-viscosity medium [LVM, 0% methylcellulose (MC) in artificial seminal fluid (ASF)], only several sperm cells escaped the cluster entanglement, causing

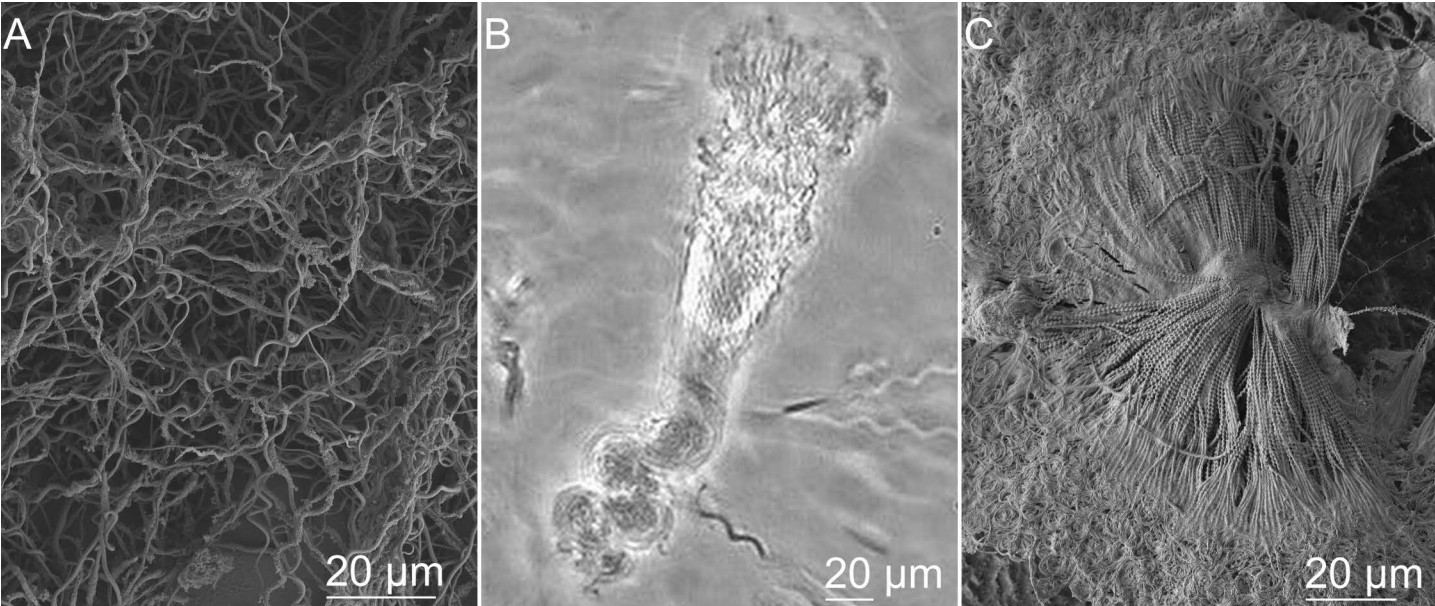

**Fig 3. Agglomeration of sperm samples.** Observed by: (A) electron microscopy in ray *Potamotrygon motoro*, demonstrating randomly distributed spermatozoa inside of spermatozeugmata; (B) phase-contrast microscopy in skate *Raja asterias* spermatozeugmata, demonstrating coiled together spermatozoa heads with synchronously moving flagella; (C) electron microscopy in shark *Scyliorhinus canicula*, demonstrating spermatozoa connected by the tip of the heads and at the midpiece region. See also S1 Video.

a very slow release of spermatozoa from clusters. The sperm cells that managed to break free from clusters, however, did not manage to swim far from them. After 20 min of observation, the spermatozeugmata conserved their structure, with most spermatozoa residing inside the cluster despite the vigorous motion of individual cells and their collective beating bundles. This scenario changes under high-viscosity fluid environments (HVM, 1–2% MC in ASF). In this case, all spermatozoa were effectively released in a few minutes. After 20 min of observation, no remaining structure of spermatozeugmata could be found in 2% MC in all three species and all tested sperm samples.

The randomly arranged spermatozeugmata of rays (Fig 3A) were observed to release sperm fully from its entangled clusters in HVM (S1a Video) but remain in the cluster in LVM. In sharks, the radial configuration of spermatozeugmata (Fig 3B) switched into a "spikey shape" arrangement in HVM due to the progressive collective motility of the spermatozoa bundle. The connection between head tips was broken, and spermatozoa sheared relative to each other towards the centre, ending with all heads pointing outwards while still stacked at the midpiece region, with the beating flagellum in the inner core, forming a perfect monopole configuration with sperm cells aligned radially [20]; S1b Video). The rate of shape changes of the sperm cluster from a symmetrical circular arrangement to a monopole configuration and the time for how long it will remain in this shape also depended on the viscosity of the media. LVM considerably suppressed any cluster rearrangement for extended periods compared with HVM. In 2% MC, spermatozeugmata was broken in the first few seconds after contact with the medium, and the sperm spiked monopole "explodes" with the sperm concentrated at the centre released and swimming in all directions, similar to fireworks (S1c Video).

Skate spermatozeugmata are made of braided sperm, coiled together by their large-diameter helical heads whose flagellar bundle self-organizes into a collective motion with waves travelling from head to flagellum tip. Despite its collective motion, spermatozeugmata with motile flagella are nonprogressive and tumble around the same region. Interestingly, sperm reverse their flagellar beat with waves travelling from tip to head, forcing the head to spin in the opposite direction, thus effectively unentangling all sperm helices from the cluster. The flagellar bundle stops entirely before the reversal in wave progression, thus provoking a backward movement. The backwards motion and reversal in the spinning direction cause the sperm helical heads to "unscrew" from the dense spermatozeugmata cluster (S1d Video). Once spermatozoa are released from nearby cells, the flagellar wave progression switches to regular beating, and all spermatozoa swim efficiently away from the cluster. Interestingly, we observed that this switch in wave-propagation could be equally utilized to unscrew helical sperm from dense structures and reverse steering to avoid obstacles, making them highly adaptative and autonomous swimmers in a variety of environments (S1e Video).

## Cytoplasmic sleeve

Another specific structure of *Elasmobranchii* sperm is the "cytoplasmic sleeve." Before sperm activation, the sleeve covers the midpiece (Fig 2D), while it is not typically present after activation and in free-swimming sperm. However, at the onset of sperm activation, some motile spermatozoa still carry their sleeves. In some cases, it is even possible to observe the progressive release of the sleeve as it slides and corkscrew down the flagellum while the sperm carefully swims away due to hydrodynamic friction acting on the sleeve (S2a Video). When the spermatozoa leave the spermatozeugmata by their own motion, most do not have this sleeve. The sleeves remain in the cluster "exit" area. An example of such a phenomenon can be observed in S2b Video, where disaggregation of skate spermatozeugmata was slow due to the relatively low viscosity of the ASF (0.125% MC), and many sleeves can be observed surrounding the cluster of beating spermatozoa.

## Sperm motility and progressive motion under different viscosity environments

After dilution in ASF, spermatozoa were released from spermatozeugmata, forming a dilute suspension in which active motility was maintained for 30 to 40 min. As expected from the sperm morphology, the flagellar beating was helical. The flagellum rotated around its swimming axis, causing the helical head to spin as sperm swam freely in the fluid. During such movement, parts of the flagellum and head were observed to go in and out of focus, demonstrating the 3D nature of swimming (see S3a Video for a skate spermatozoon in 2% HVM). Interestingly, the spermatozoa of all three species in the LVM were almost nonprogressive, tumbling and spinning around with high frequency (velocity less than 10 μm/s), though ineffectively, and with highly asymmetric flagellar waveforms containing coiled parts on the distal ends of their flagella, restricting proper measurements of flagellar parameters at 0% MC solution (see ray sperm example in LVM in S3b Video). The sperm motility efficiency increased with increasing viscosity (Fig 4). At the same time, in such conditions, the linear velocity is strongly connected to the sperm's ability to spin around its longitudinal axis, effectively "corkscrewing" the sperm into the medium. The highest progressive velocity for shark sperm was 31.3 ± 1.9 μm/s observed at 0.75% MC (Fig 4B and S3c Video). In comparison, 1% MC instigated higher speeds in ray 51.6 ± 12.9 μm/s and skate sperm 64.8 ± 2.9 μm/s (Fig 4B and S3d, S3e Video).

## Flagellar wave parameters under different viscosity environments

As seen in Figs 4A and 5, the flagellar shape is highly affected by viscosity. The increased viscosity of the fluid limits the propagation of flagella beating from side to side, decreasing its overall amplitude (from 9.1 ± 0.7 μm at 0.75% MC to 3.9 ± 0.4 μm at 2% MC in shark; from 9.6 ± 0.7 μm at 0.75% MC to 4.7 ± 1.4 μm at 2% MC in skate) and wavelength (from 36 ± 1 μm at 0.75% MC to 26.5 ± 1.8 μm at 2% MC in shark; from 30.5 ± 1.8 μm at 0.75% MC to 22 ± 1.4 μm at 2% MC in skate). Exceptionally, the amplitude of ray sperm flagella seemed less affected by the increased viscosity. However, bending waves shifted axially due to the wave compression phenomenon [15], slightly decreasing

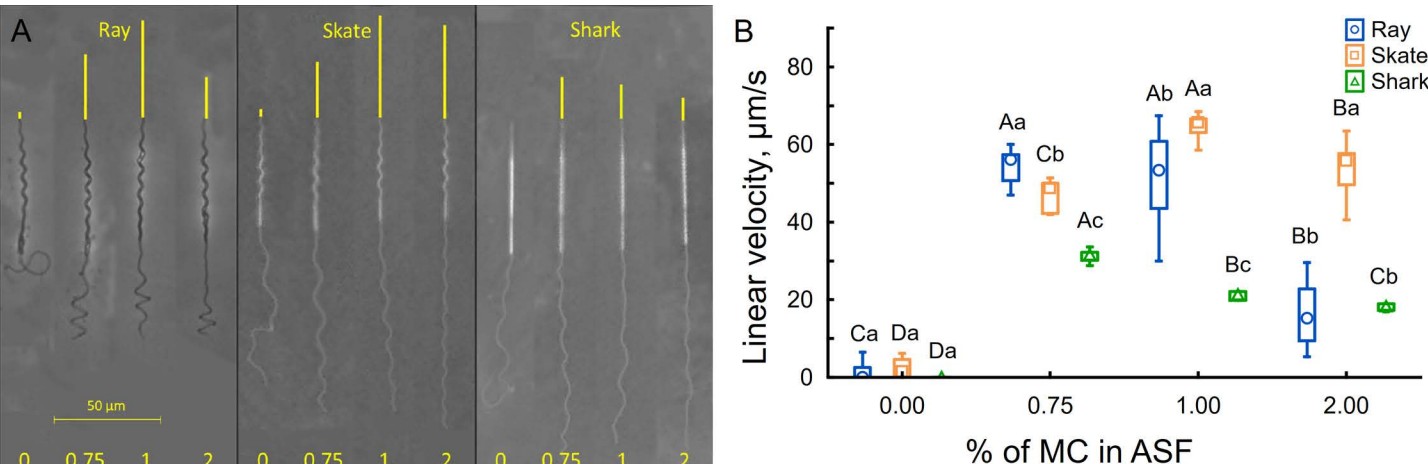

**Fig 4. The velocity of ray *Potamotrygon motoro*, skate *Raja asterias*, and shark *Scyliorhinus canicula* spermatozoa at different viscosities (0, 0.75, 1, 2% MC present in ASF).** (A) - example of flagellar shape and passed the distance of spermatozoa; (B) – linear velocity. Each experimental group has an average of 8 (3–15) measured spermatozoa. Significant differences inside the same species are marked by capital letters, and differences inside one viscosity concentration are marked by lowercase letters (nonparametric Kruskal-Wallis ANOVA followed by multiple comparisons of means ranks for all groups, p < 0.05). See also S3 Video.

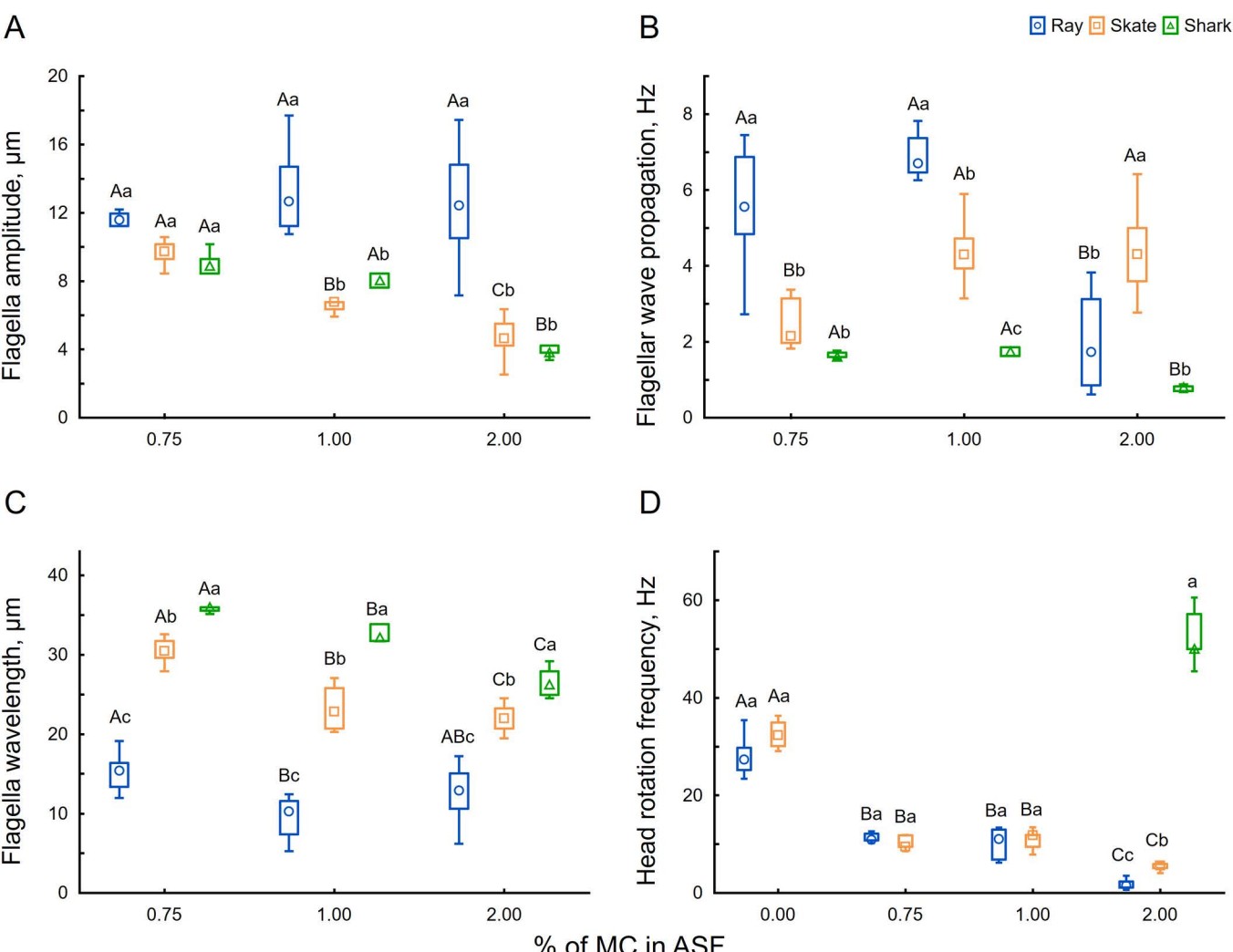

**Fig 5. Parameters of spermatozoa and flagella motility of ray *Potamotrygon motoro*, skate *Raja asterias*, and shark *Scyliorhinus canicula* spermatozoa at different viscosities (0, 0.75, 1, 2% MC present in ASF).** (A) – flagella amplitude; (B) – flagellar wave propagation frequency; (C) – flagellar wavelength; (D) – head rotation frequency. Data are presented as median, 75, and 25 percentiles (box), min, and max values (whiskers). Each experimental group has an average of 8 (3–15) measured spermatozoa. Significant differences inside the same species are marked by capital letters, and differences inside one viscosity concentration are marked by lowercase letters (nonparametric Kruskal-Wallis ANOVA followed by multiple comparisons of means ranks for all groups, p < 0.05).

wavelength without a visible effect on flagellar amplitude (Figs 4A and 5A, 5C). Thus, we can not observe a significant correlation between wavelength and tail amplitude of ray spermatozoa at different viscosity as presented in Fig 6A, in contrast to a significant correlation for skate and shark spermatozoa (Fig 6B, 6C). Moreover, partial wave compression was also observed in skate and shark spermatozoa when they were more compressed and shifted to the distal end of the flagellum. The part of the flagellum proximate to the midpiece remains active during such motion and oscillates with extremely high frequency and almost undetectable amplitude (S3f Video).

## Helical head spinning

When the flagellum propagates helically, the whole spermatozoon spins around its longitude axis. Thus, the rotation frequency could be related to sperm velocity, especially given their

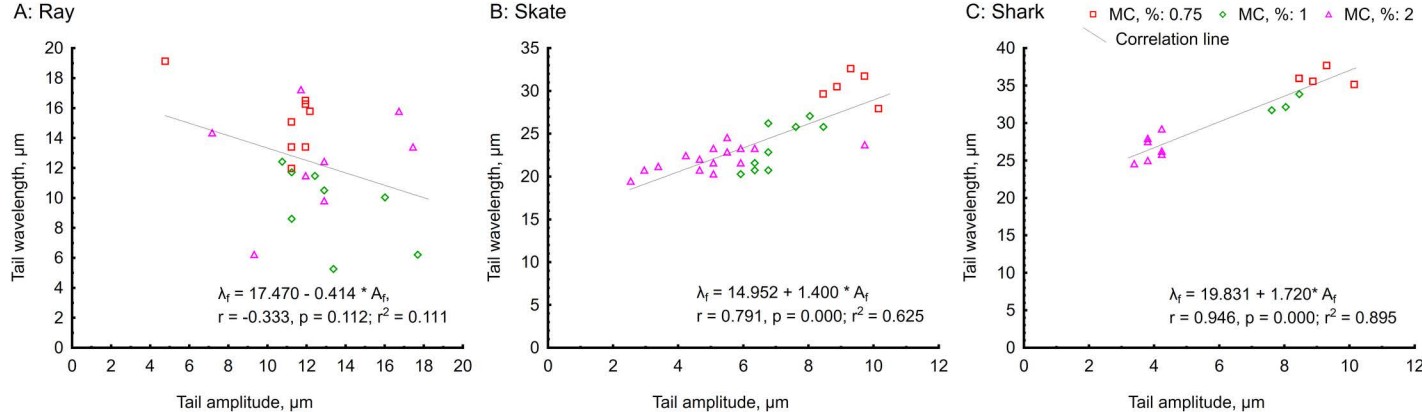

**Fig 6. Correlations between tail wavelength and amplitude under conditions of different viscosity (0, 0.75, 1, 2% MC present in ASF).** (A) ray *Potamotrygon motoro;* (B) skate *Raja asterias*, and (C) shark *Scyliorhinus canicula* spermatozoa. Each individual point on a graph represents measurements of a single spermatozoon. Regression equation, r, r2, and P-values are presented in the bottom right corner of each graph.

helical sperm architecture. In our study, the observed frequency of flagellar propagation was low and sometimes almost identical to rotation frequency, up to 6 Hz for all three species and studied viscosities (Fig 5B). In the condition of LVM, the frequency of head spinning was very high, reaching rates higher than 50 Hz, but usually with a frequency between 20 and 35 Hz for skate and ray. Increasing media viscosity stabilized rotation frequency at approximately 10 Hz for skate and ray spermatozoa (Fig 5D). The correlation of velocity and total frequency of a head rotation and flagellar wave progression under different viscosity is presented in Fig 7. The head-spinning frequency in sharks was only possible to measure at 2% MC, demonstrating exceptionally high rates, around 50 Hz (Fig 5D).

## Steering long sperm helical bodies in high-viscosity

During the regular motion of such a long, spinning helical head, *Elasmobranchii* spermatozoa usually swim in a straight line. However, we observed that steering their long helical heads was

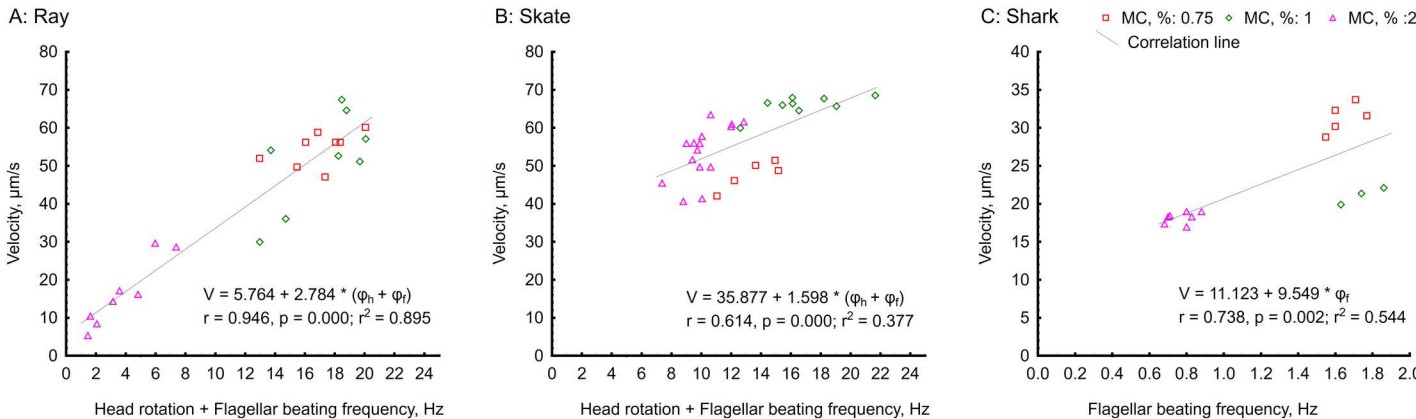

**Fig 7. Correlations between velocity and total frequency of a head rotation and flagellar wave progression under conditions of different viscosity (0, 0.75, 1, 2% MC present in ASF).** (A) ray *Potamotrygon motoro*; (B) skate *Raja asterias*; and (C) shark *Scyliorhinus canicula* spermatozoa (in the case of shark, only the frequency of flagellar wave propagation is present). Each individual point on a graph represents measurements of a single spermatozoon. Regression equation, r, r2, and P-values are presented in the bottom right corner of each graph.

possible in high-viscosity media through forced buckling of the elastic head. Depending on the level of helical buckling, the swimming trajectories ranged from smooth turns (S4a Video) to sharp bends ending in large directional changes (Fig 8 and S4b Video).

This helical buckling is unique because while it occurs, the helical head continues to rotate around its longitudinal axis with a high spinning frequency and a conserved spinning direction (S4b Video). The head buckling could also be transient, leading to fast and sharp temporal directional changes (S4c Video). No head helical buckling could be observed in LVM. In such LVM situations, we only observed the head bending after the tail entangled the spermatozoa head during backward wave propagation in ray spermatozoa (since it could not propagate and rotate in one place). During this process, the spermatozoa head was bent, and after a while, the flagellum was released, leaving the head free again. When the head was not held by the flagellum anymore, its shape immediately returned to its unstressed straight configuration (S4d Video).

The backward motion of spermatozoa, which could be used to disentangle sperm from spermatozeugmata, was also utilized to change the swimming direction and steer spermatozoa. This swimming reversal re-aligns the long helical sperm head to avoid obstacles or penetrate a more viscous environment (S4e Video).

After release from spermatozeugmata, the spermatozoa moved linearly away from the centre until they reached the edges of the viscous drop, where the properties of media differ (due to evaporation, surface tension, etc.). Precisely at this moment, near the drop boundary, the swimming direction of most of the cells changed, and the process of sperm steering was observed. An example of the different stages of sperm release, the linear progressive motility, and the sperm steering as the drop boundaries are approached can be found in S4f Video.

## Discussion

Our results shed light on the fundamental principles of sperm motility in *Elasmobranchii.* Their unique helical head morphology and three-dimensional flagellar beating enable sperm cells to move in a highly viscous environment. We show that in low viscosity, sperm cells are nonprogressive while gaining the ability to move progressively in an environment with high viscosity. High viscosity was required during spermatozeugmata dissociation, allowing spermatozoa to unbundle from the others and move progressively and straightforwardly. This strongly indicates that low-viscosity environments suppress sperm progressive motility and release from spermatozeugmata in these species. Additionally, increased viscosity also creates

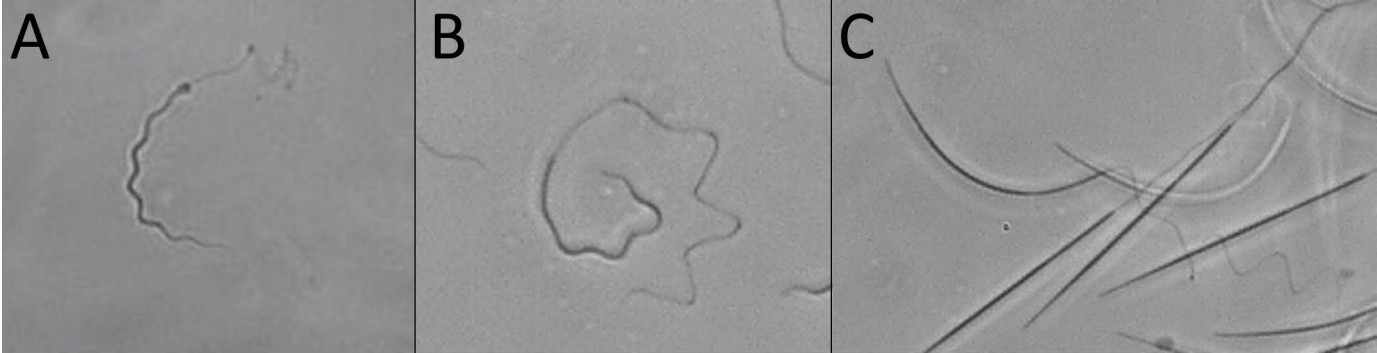

**Fig 8. Pictures of motile spermatozoa in 2% MC demonstrate buckling of the head, leading to a sperm turning.** (A) ray *Potamotrygon motoro*; (B) skate *Raja asterias*, and (C) shark *Scyliorhinus canicula*. See also S4 Video.

an environment under which elongated helical heads of spermatozoa can buckle during motion, thus enabling directional change critical for navigation purposes.

The spermatozoa of *Elasmobranchii* possess specific characteristics (apomorphies) such as elongated heads (bigger than most internal and external fertilizers [21], elongated midpieces, longitude columns, and cytoplasmic sleeves [22]). These lead to distinct elastohydrodynamic effects compared to microswimmers with smaller sizes and simplified structures. The elasto-hydrodynamic sperm number parameter [15,23] contrasts flagellar bending and viscous forces acting on the tail and is proportional to the flagellar length $L$, where the viscosity of the fluid is denoted by $\eta$, frequency of the beat is $\omega$, and the bending stiffness of the flagellum is $E_b$:

$$Sp = L\left(\eta\omega / E_b\right)^{\frac{1}{4}}$$

Low Sp values are associated with effectively "stiff" tails, while large Sp is linked with effectively "floppy" behaviour of the flagellum. Having large Sp, due to a very long flagellum or high viscosity of the fluid, poses complex challenges to be overcome by the sperm, as it makes it harder to propagate waves due to the large elastohydrodynamic dissipation continually acting along the flagellum [24].

Large sperm number Sp also makes the flagellum prone to flagellar buckling instability [15] and nonprogressive swimming [25]. Interestingly, mammalian species overcome these difficulties by reinforcing the flagellum with large ultrastructural elements that taper along the flagellum to prevent flagellar buckling and promote sperm progressive motility in high Sp regime [15,26]. *Elasmobranchii* species, however, recur the longitudinal columns but distribute them helically along the flagellum, forcing the flagellar bending into a helical shape during the beating. Hence, instead of pushing the long head by the beating flagellum, the flagellum rotates around the swimming axis to induce torque around the elongated helical head – the helical shape stabilizes the rotation around the head axis [27]. Ultimately, this generates the exquisite corkscrewing motion of the helical head into the thick fluid environment. However, this mechanism was ineffective in our experiments under low-viscosity environments when Sp is low. Overall, this avoids flagellar buckling and transfers most of the motion in the direction which seems more effective for this size and shape of the cell: the rotation of the helical head. Altogether, the body-to-flagellum helical architectures allow a matched helical motion, with rotation of the flagellum inducing the corkscrewing of the large-sized head into the media, making swimming more effective in highly viscous fluids. Indeed, our results demonstrated that despite the specific shape and large sizes of the *Elasmobranchii* spermatozoa, they could reach swimming speeds similar to the much smaller mammalian spermatozoa in a similar medium, as an example of human spermatozoa with an average of 62 µm/s of progressive velocity [28].

The above-described physical background for differences in spermatozoa swimming with differentially shaped heads is essential for understanding fundamental relationships between spermatozoon shape and size evolution and its biological performances [21]. Moreover, it is suggested that spermatozoa physiology is a complex subject that should be studied, considering the balance between reproduction strategy, specific physiological environment (conditions of fertilization), and the ability of sperm to adapt, progress, and react to the changes under this environment. Several studies attempted to correlate sperm size and motility in different species, including fish, mammals, birds, etc [29–33], including cartilaginous fish [5,18], sometimes reporting no or even negative correlation [34–37]. Most of the studies simplified spermatozoa size to a single parameter, such as flagellar length or whole spermatozoa length, without considering the head size or shape or not optimizing the swimming environment.

This could explain the high heterogeneity of results and sometimes contradictory conclusions reported in the literature.

This study aimed to test the hypothesis that the head-to-flagellum shape of *Elasmobranchii* spermatozoa is specifically adapted to high-viscosity environments, as found in physiological conditions. To investigate this, we evaluated the sperm performance of three *Elasmobranchii* species across different physiological scenarios involving the release from spermatozeugmata, the progressive motion, and the regulation of motility.

## Activation of spermatozoa

All *Elasmobranchii* species are internal fertilizers. At the same time, the fertilization process itself is not fully understood in these species. Males deposit spermatozoa through specific grooves on claspers (extended pelvic fins) during mating [2]. However, these channels are not fully closed, allowing contact with the aquatic environment (seawater in marine species and freshwater in freshwater species). Also, the sperm is propelled into the female by the accumulation and subsequent release of water from the siphonal sacs in sharks [38,39] or, presumably, by secretions from claspers glands in rays and skates [40]. Thus, sperm motility could potentially be activated by external factors similar to external fertilizers [41]. On the other hand, the unique physiology of marine *Elasmobranchii* species sets them apart from other vertebrates since their blood and seminal plasma osmolality is approximately 1000 mOsm/kg (equivalent to seawater), thus preventing osmotic shock during transition. In contrast, freshwater *Elasmobranchii* have internal body fluid osmolality ranging between 243–350 mOsm/kg. In this case, contact of spermatozoa with freshwater induces osmotic shock, potentially resembling signalling in freshwater external fertilizers. However, in our previous observations, freshwater led to a rapid cessation of flagellar beating [42]. These findings suggest that the motility of *Elasmobranchii* spermatozoa is not directly regulated by environmental osmolality. Nevertheless, further research is needed to explore the potential effects of the siphonal sac or clasper gland fluids on sperm physiology during mating and to uncover the precise mechanisms involved in sperm deposition and interaction with the female's internal environment. Our observations indicate that the spermatozoa flagellum remained actively beating in undiluted seminal fluid and ASF across all three species, even when spermatozoa were still compacted within the spermatozeugmata. When LVM was used, the spermatozoa were nonprogressive, tumbling in place, and struggled to leave the spermatozeugmata unless it was mechanically disrupted during mixing with ASF. Interestingly, an increase in ASF viscosity was sufficient to trigger progressive sperm motility. Thus, we suggested that during mating, the already activated spermatozoa (still packed in spermatozeugmata) are transported towards the female's internal organs, where the next phase of progressive motility is initiated upon encountering the more viscous internal environment.

## Spermatozeugmata and spermatophores

Similar aggregations of spermatozoa, when spermatozoa connected without surrounding matrix (unencapsulated), were observed in different *Elasmobranchii* species and called spermatozeugmata [43–46]. Clumped spermatozoa were also observed surrounded by additional structures (matrix), thus being embedded inside capsules or spermatophores and observed in several other *Elasmobranchii* species [47,48].

Several authors suggest that *Elasmobranchii* species may store spermatozoa after ejaculation in the terminal zone of the oviducal gland, ensuring a supply of sperm for successive fertilization of ova released during ovulation over a period of several weeks or months [49–51]. Thus, the presence of spermatozeugmata or spermatophores in ejaculation could

have several biological functions. It can keep spermatozoa aggregated during transport along the female reproductive tract if the water flow from the siphonal sacs carries sperm close to the fertilization site (presumably, the oviducal gland). In this case, the aggregations would not disintegrate, as the surrounding medium (water) would be low viscosity. Another possible role can be associated with enhanced motility of the entire group of spermatozoa. This has been suggested for the clusters of spermatozoa that appear in some monotremes and apparently are similar to rays [11]. Last, these aggregations may be linked to sperm competition between males in polyandrous species where multiple matings have been described as an example of elasmobranchs [52]. In this case, the formation of sperm bundles has been considered an adaptive mechanism to improve the reproductive fitness of males due to selective pressure to optimize their sperm delivery. In this way, sperm can arrive more quickly at the storage area or more easily displace sperm from other males [11,53].

Spermatozeugmata and spermatophores could be stored safely inside the female body for a long time, waiting for a signal to release and start a free active movement to ensure fertilization at the right time. That mechanism could be similar to the synchronization mechanism in external fertilizers when sperm is activated only after ejaculation and contact with water [54]. Still, the elements of such a signalling cascade should be further discovered. In some mammals, fertilization synchronization is realized through the capacitation process when only a small part of the sperm population is capacitated and ready to fertilize during the ovulation period, thus prolonging the time window for successful fertilization [55], but for fishes, sperm capacitation was described only in a few species [56,57]. For many Chondrichthyan species, synchronization may be needed to ensure fertilization immediately after ovulation and before egg encapsulation [2]. At the same time, the process of spermatozoa releasing from agglomerations and capsulation of the eggs in *Elasmobranchii* has not been studied in detail. In our study, the process of sperm unbundling from spermatozeugmata was significantly accelerated when the HVM was used for sperm dilution, resulting in a burst of spermatozoa motility in all three studied species. Thus, the increased viscosity may play a role in the trigger of sperm unbundle in these species.

We observed, specifically for *Elasmobranchii* species, cytoplasmic sleeves structure presented in all of the studied species, which was also reported for *Scyliorhinus canicula* previously [58]. From our results, we can not conclude the exact function of these sleeves and if they are involved in spermatogenesis, playing the Hermes body's role, as suggested before [47]. The Hermes body, or the cytoplasmic droplet, is a component of the sperm flagellum unique to epididymal spermatozoa, originating as the remnant of germ cell cytoplasm during spermatogenesis. Its precise role and function are unclear, but several authors hypothesize that this structure might be associated with the maturation process and acquisition of sperm motility [59,60]. Alternatively, the observed sleeves could be reminiscent of the structural matrix that facilitates the interaction between Sertoli cells and developing spermatozoa during spermatogenesis since they are not a part of the spermatozoa itself but rather provide an additional layer or cover at the midpiece with this extra membrane. Our results confirm that the sleeves are not required for proper motility of spermatozoa. Instead, it could be associated with the unbundling process since, commonly, spermatozoa remove them while exiting the spermatozeugmata. Thus, we suggest that the cytoplasmic sleeves connect spermatozoa inside spermatozeugmata. In that case, the spermatozoon can rotate around its axis inside the sleeve (like inside a tube) while still connecting to other spermatozoa inside the spermatozeugmata.

It should be mentioned that some of the authors reported sperm storage at the oviducal gland involving only individual spermatozoa, which are basically stacked inside the gland tissue in the terminal zone [61]. Currently, there is no direct indication of how exactly sperm storage is ongoing. Many questions still exist and require further study and deeper

consideration: Do the spermatophores or spermatozeugmata preserve their structure during storage inside the female reproductive tract? What are the exact triggers of spermatozoa activity and unbinding process in species with different sperm aggregation structures? Are the cytoplasmic sleeves involved in organizing spermatozoa inside spermatozeugmata or spermatophores, and what is their exact function? How do spermatozoa aggregates impact sperm competition and cryptic female choice in different species? Those questions are fundamental for a general understanding of the reproduction strategy. They should be further considered, taking into account specific spermatozoa physiology and their ability to progress and adapt to specific environmental viscosity.

## Sperm progressive motility

After being released from spermatozeugmata, nearly 100% of spermatozoa moved straightforwardly in HVM. However, spermatozoa stayed almost nonprogressive in LWM, even though the flagellum is active and head rotation frequency is high. In this situation, the rotating spermatozoa head could move the liquid around without effective propulsion, while flagellar waves did not produce enough force to push forward cells with such a large helical head. When the medium viscosity increased, the situation changed. Instead of rotation without propulsion, the spermatozoon head began to screw into the media with the help of the helical shape head and the helical flagellar wave, thus progressing forward. We could observe high viscosity effects on the flagellum: more symmetrical flagella wave and decreasing wave amplitude and wavelength as described previously [28]. Consequently, spermatozoa began to swim more straightforwardly since the side-to-side movement of the head and flagellum was limited by increased viscosity. Nevertheless, *Elasmobranchii* spermatozoa (particularly ray spermatozoa) were prone to wave compression [25,62]. The lateral to midpiece part of the flagellum remained highly active, producing high-frequency shallow amplitude waves. This behaviour suggests that different parts of the flagellum could be regulated differently, depending on the environment, leading to the appearance of varying shapes of waves along the flagellum. Surprisingly, the sperm motility in high viscosity was not hindered by the fact the amplitude of the flagellar waves was very small. It may be attributed to the induced head spinning by the flagellum. This suggests that sperm penetration in high viscosity can be achieved with two distinct modes in *Elasmobranchii*: the classical way in which large amplitude bending waves propel the large head forwards and a novel mechanism in which bending waves are insufficient to produce cell propulsion but sufficient to induce torques on the helical head that subsequently corkscrew into the medium.

The recent publication of Wang et al. [14] suggested that head shape could contribute up to 31% efficiency of forces used for spermatozoon propagation/movement. The authors tested the Heterogeneous Dual Helical Propulsion Mechanism, explaining the high adaptability of ray spermatozoa to different viscosities. In many aspects, the observed spermatozoa behaviour that they described, including the rotational basis of sperm progressive motion, rotation frequency, and velocity of spermatozoa, are very consistent with our observation, except for the effect of viscosity on spermatozoa progressive motility. In this case, the most efficient spermatozoa motility was observed in LVM. Our study observed the highest velocity for shark sperm at 0.75% MC, at 1% MC for ray and skate, and a slight decrease in higher viscosities. This difference could be related to species-specific differences in motility regulation or the differences in the chemical effect of compounds used to create viscosity (MC in our study and alginate in Wang et al. study) and will require further investigations. Interestingly, Wang et al. [14] observed similar tail deformation in HVM due to the curling of the flagellar tip and entangling of the flagellum on the helical head, a typical observation in our study for ray spermatozoa in

LVM. Critically, in HVM, we found that close to 100% of observed spermatozoa were progressive, demonstrating efficient forward motion with highly symmetrical flagellar waves.

As discussed above, the properties of spermatozoa observed in our study for *Elasmobranchii* could be easily applied to explain sperm motility in any other species with similarly shaped spermatozoa as an adaptation to high viscosity during fertilization, for example, birds and amphibians [63,64]. This suggests that environmental viscosity is a highly overlooked parameter affecting sperm behaviour in many species. Increased drug forces in high viscosity may provoke adaptation by elongating the head and applying a helical structure for the rotational mode of progressive motility in large-sized evolutionarily conserved sperm types. At the same time, viscosity could be one of the driving forces for further adaptions, leading to decreasing spermatozoa size and simplifying spermatozoa structures, which could be observed in many modern species (as examples of numerous mammals and teleost fishes).

### Sperm navigation/directional changes

After release and finding optimal conditions for progressive motion, all spermatozoa continue to move straightforwardly, with the primary objective of locating and fertilizing the egg. Recently, the common hypothesis of random fertilization has been replaced with the guidance hypothesis [65]. According to this hypothesis, spermatozoa sense the changes in the environment and adapt their motility accordingly, thus increasing the chance of encountering/meeting the egg. If the cells can control motility dynamically (time) and spatially (navigation), the opportunities for fertilization will be highly improved. Thus, the ability of spermatozoa to react to environmental changes may predetermine the fertilization success and increase the value of such males, providing additional selection criteria. Our study showed the sperm's ability to steer and change direction successfully despite the large sizes and complex motility mechanism. This is done by uniquely exploiting the head buckling in high viscosity due to elastic buckling instability [15,23,66]. Simpler buckling phenomena have been explored by bacterial hook flagellum to induce directional changes in navigation [67].

In our study, in addition to the already observed and discussed phenomenon of the backward motion of spermatozoa [47,68], we also demonstrated that this is involved in sperm release from the bundle. Moreover, we observed that these long helical spermatozoa could exploit a head-to-tail buckling phenomenon to change swimming direction, demonstrating a sophisticated strategy compared to simpler microorganisms with smaller bodies [67]. This mechanism is probably passive and may appear in a situation when high forces coming from the flagellum (relatively high flagellar amplitude) push the spermatozoon head forward while spinning (screwing) speed slows down. The same spermatozoon can quickly change the bending/turning direction and thus could respond rapidly to changes in the properties of the surrounding medium. Interestingly, the media can not keep the buckled head's shape in LVM conditions due to the fast relaxation time associated with an effectively "stiff" passive elastic rod [69], equivalent to a low Sp regime for an active flagellum (as discussed above). This indicates the elastic helical head did not possess any intrinsic internal moment or pre-deformed configuration. The helical head structure thus appears to respond as a passive elastic helical rod.

### Conclusions

We investigated the spermatozoa of three *Elasmobranchii* species, the presence of spermatozeugmata, the bundle formation and unbundling processes, their progressive motility, and directional changes in navigation. Environmental viscosity was key in all aspects of motion, and specific spermatozoa structures were used to allow motion in high-viscosity conditions.

Our results suggest that these cells perform optimally in high-viscosity media and should be considered for future spermatological studies and possible artificial reproduction in these essential species. The observed helical head-to-flagellum architecture in these spermatozoa may suggest that high viscosity is one of the main environmental conditions affecting and shaping the spermatozoa and their performance during evolution to the state in which we observe them now.

## Materials and methods

### Experimental models and ethics approval

Three *Elasmobranchii* species were used in this study: *Scyliorhinus canicula* (n = 4), *Raja asterias* (n = 4), and *Potamotrygon motoro* (n = 3). The small-spotted catsharks *Scyliorhinus canicula* were randomly selected from a sample of 11 male individuals sharing their aquarium with no other species. The sharks were kept in an 8,000 L aquarium with recirculating seawater (temperature: 16–18°C; salinity: 35–37‰) and fed twice daily with herring, squid, and shrimps. All animals were adult males of an estimated age of 5 years. The maturity status was determined by assessing the degree of calcification of the claspers (a method previously employed in other studies on sperm quality). Sperm samples were collected from sharks without using anaesthesia, turning them down to get tonic immobility. Manipulation of animals was approved by the Oceanogràfic Animal Care & Welfare Committee at the Fundación Oceanogràfic Valencia, Spain (Project reference: OCE-16–19).

The Mediterranean starry skate *Raja asterias* specimens came from commercial fisheries. The animals, already dead at the time of collection, were obtained fresh from the fish market approximately five hours after capture and kept on ice until pick-up. The appropriateness of sperm samples collected post-mortem use was proved in previous studies [70]. The animals came from artisanal trammel net fisheries in the Gulf of Valencia, Spain. All animals were adult males of undetermined age. Maturity status could be determined by the degree of calcification of the claspers and the development of testes and epididymis.

The ocellate river stingray (*Potamotrygon motoro*) was kept in experimental facilities at the Faculty of Fisheries and Protection of Waters, University of South Bohemia in České Budějovice. These facilities are certified by the Ministry of Agriculture of the Czech Republic for breeding potamotrygonids and using them as experimental animals (reference numbers: 56665/2016-MZE-17214 and 55187/2016-MZE-17214). The stingrays were housed in a 900 L aquarium with recirculating freshwater, maintained at 25–26°C and 100% oxygen saturation, with a pH range of 7.2–7.4. They were fed twice daily with dry pellets and frozen forage fish. All the animals used in the study were adult males aged between 5 and 7 years. Sperm samples were collected following immersion anaesthesia in 100 mg L$^{-1}$ tricaine methanesulfonate (MS 222) and 200 mg L$^{-1}$ NaHCO$_3$ [71].

All animal manipulation procedures were conducted in accordance with the Animal Research Committee of the Faculty of Fisheries and Protection of Waters, following the principles based on the EU-harmonized animal welfare act of the Czech Republic and the principles of laboratory animal care in compliance with the national law (Act No. 246/1992 on the protection of animals against cruelty).

### Sample collection and sperm motility video recordings

Sperm samples were collected from a shark - *Scyliorhinus canicula* (n = 4), skate - *Raja asterias (*n = 4), and ray species - *Potamotrygon motoro* (n = 3) males after abdomen massage according to recommendations for cartilaginous fishes [72,73] and stored at 4 °C before motility recordings. One aliquot of sperm (0.1–0.3 µl) was deposited in 40 µl of artificial

seminal fluid (ASF) on a glass slide using the tip of an injection needle for further observations and recording. To observe spermatozeugmata and the process of sperm releasing, the raw sperm samples were carefully placed in the middle of ASF flat drop under the microscope without mixing for each sperm sample (preserving in this way the natural physiological condition by which sperm is found in ejaculate). One observation field (with 1–3 spermatozeugmata) was followed for 20 min to estimate the efficiency of spermatozoa release. To observe spermatozoa realizing and progression towards the boundary of the drop, we moved the observation field following motile spermatozoa rather than focusing on spermatozeugmata location. To estimate motility duration, sperm aliquots (0.5–1 μl) were diluted with 0.5 ml of ASF in Eppendorf, and the presence of active (moving) flagellar was checked every 10 min under the microscope. The composition of the ASF was formulated to mimic its natural seminal fluid osmolality and ionic composition in the target species. For the freshwater species, *P. motoro,* ASF consisted of: NaCl 130 mM, KCl 8 mM, $CaCl_2$ 0.6 mM, Glucose 0.4 mM, Tris 10 mM, pH 7.84, osmolality 295 mOsm/kg modified after [56]. For the marine species, *S. canicula* and *R. asterias*, ASF consisted of: urea 433 mM, NaCl 376 mM, Trimethylamine N-oxide (TMAO) 120 mM, KCl 8.4 mM, Glucose 50 mM, $CaCl_2$ 7 mM, $NaHCO_3$ 3.5 mM, $Na_2SO_4$ 0.08 mM, 1.4 mM $MgSO_4$, adjusted to pH 6.5 and osmolality 1000 mOsm/kg [73]. To vary the viscosity of ASF and cover the physiological diapason of internal fluids, the concentrations of 0, 0.125, 0.5, 0.75, 1, and 2% of Methylcellulose (MC; Sigma-Aldrich, M0512) were added [16]. These solutions have correspondingly resulted in approximate viscosity of 0, 5, 25, 80, 200 and 4000 mPa × s (according to Methylcellulose M0512 manufacturer product information). The solutions with 0% MC are considered low viscosity (LVM), solutions with MC concentrations of 0.75% or above are considered high viscosity (HVM), while 0.125 and 0.5% are considered medium.

The motility of all three species (11 males in total) was recorded under room temperature conditions (22–24 °C) during the initial 2–5 minutes after placing sperm in ASF. Records were done with a digital video camera (IDS Imaging Development Systems GmbH, Obersulm, Germany) set to 193 FPS (800 × 600) and a high-speed video camera (Olympus i-speed TR, Tokyo, Japan, providing 848 × 688 pixels spatial resolution) at 2000 FPS mounted on a phase-contrast microscope and x20 or x100 objectives for several minutes (depends on experimental conditions). Motility records were stored in AVI format before analyses.

### Analysis of sperm motility and flagella characteristics

Standard CASA methodology is inadequate for analyzing the motility of these specific spermatozoa and does not yield valuable data to comprehend complex motions due to difficulties with tracking long (non-spherical) heads and due to the three-dimensional nature of sperm motility, relying on rotation. Therefore, we have developed a bespoke method to observe and process this unique motility. Our approach lets us directly capture flagellar and head motion, manually extracting information about motility and rotation. Digital frame-to-frame analyses of our video-microscopy recording were performed to measure linear swimming velocity, flagellar waveform amplitude, wavelength and frequency of wave oscillation, and sperm head-spinning frequency in ImageJ software (U. S. National Institutes of Health, Bethesda, Maryland, USA). Since the whole spermatozoa rotate during the movement, the actual frequency of flagellar oscillation is a sum of both observed wave propagation frequency plus the frequency of spermatozoa rotation. If the wave propagation is not detected at all ("frozen" wave of flagellum), it will mean that the frequency of flagellar propagation is equal to the frequency of rotation. The detection of rotation in shark sperm was challenging due to the small radius of the head helix being partly out of focus in lower

viscosities, also restricting the number of measurements for shark spermatozoa in 1 and 0.75% MC to 4 and 3 correspondingly, and was only possible to measure at 2% MC at x100. An example of measured parameters is shown in S1 Fig, where the sperm head-spinning frequency is calculated directly from the spinning motion of the helical-shaped head, which travels along the head axis, similarly to a propagating wave.

## Analysis of spermatozoa head morphology and electron microscopy

To visualize the precise shape and size of the spermatozoa used in this study, we estimated morphometry parameters from video recordings and performed TEM on fixed samples.

Frames obtained from video records were processed to measure head length, head helix amplitude, and wavelength, as presented in S2 Fig. The number of helices is calculated as the total number of all crests on both sides of the head divided by 2.

Sperm samples were fixed with 2.5% glutaraldehyde in 0.1 M phosphate buffer for two days at 4 °C. For Transmission Electron Microscopy (TEM), samples were post-fixed in osmium tetroxide for 2 h at 4 °C, washed, dehydrated through an acetone series, and embedded in resin (Poly/Bed 812; Polysciences, Inc., Warrington, USA). A series of ultrathin sections were cut using a Leica UCT ultramicrotome (Leica Microsystems, Wetzlar, Germany) after being counterstained with uranyl acetate and lead citrate and examined in a TEM JEOL 1010. The fixed samples for the Scanning Electron Microscope (SEM) were step-wise dehydrated in an acetone concentration, dried using a critical point dryer PELCO CPD 2 (Ted Pella Inc., California, USA), and coated with gold in vacuum SEM Coating Unit E5100 (Polaron Equipment Ltd., California, USA). Samples were examined with an SEM JSM 401-F or SEM JEOL 6300 (JEOL Ltd., Tokyo, Japan) equipped with a Sony CCD camera.

## Quantification and statistical analysis

Due to the three-dimensional nature of sperm motility, our analysis focused on spermatozoa where observable head and flagellar motions consistently appeared within the field of observation and focus, thus restricting our measurement numbers. In total, 106 spermatozoa were analyzed from 11 different males. Depending on experimental conditions, each experimental group included an average of 8 spermatozoa (ranging from 3 to 15) that were measured and analyzed. The data set with all the measurements per individual spermatozoa is presented in the S1 Table. The parameters of the sperm velocity and flagellar parameters are presented in the text as mean ± standard deviation. The limited sample size for assessing sperm motility and flagellum characteristics, non-normally distributed values inside of these experimental groups ($p < 0.05$ in Kolmogorov-Smirnov test), and no homogeneous variances ($p < 0.05$, Levene's test) suggested application of nonparametric statistical analysis. Thus, the nonparametric Kruskal-Wallis ANOVA followed by multiple comparisons of means ranks for all groups was performed using Statistica (version 13, TIBCO Software Inc., 2017, Palo Alto, CA, USA). Statistically significant differences in values inside the same species are marked by capital letters, while differences inside one viscosity concentration are marked by lowercase letters in corresponding figures.

The parameters of the sperm head are presented in the text as mean ± standard deviation from data obtained from 18 spermatozoa of rays (3 males; 5, 6 and 7 spermatozoa per male), 15 spermatozoa of skates (4 males; 5, 3, 2, 5 per male) and 23 spermatozoa of sharks (4 males; 8, 5, 5, 5 per male). The analysis was performed using the same spermatozoa frames used for motility estimation, which allowed accurate measurements. Correlation analysis was performed using Statistica (version 13, TIBCO Software Inc., 2017, Palo Alto, CA, USA) to

estimate Pearson Correlation (r), r², P-value and to determine a linear regression equation presented in the figures.

## Supporting information

**S1 Table. The data set.** Includes all the measurements per individual spermatozoa.
(XLSX)

**S1 Fig. Example of skate flagellar and head motility analysis.** The black line corresponds to the line connecting the tips of the head on each frame. The yellow and orange lines connect the corresponding "waves" on the spermatozoa head, and the green and red connect waves on the tails. By calculating intervals (number of frames) of the line of the same colour crossing the black line, we can estimate the time needed for one complete rotation or beat cycle (0.05 s between each frame). The number of such cycles per second will be expressed as frequency (Hz). The distance between the green and red lines provides information about the length of the flagellar wave (μm). Recalculating the distance spermatozoa tip travelled during one second will give us info about sperm velocity V (μm/s). The amplitude of the flagella wave is the distance between two parallel lines connecting the waves of the flagellum from both sides, A (μm). Related to the Methods section.
(TIF)

**S2 Fig. Example of skate head morphology measurements.** The distance from the tip of the head to the beginning of the flagellar is measured as head length (including midpiece). The distance between the two parallel lines connecting the waves of the head helix from both sides is the amplitude of the head helix. The length of the head helix is measured as the average distance between two wave crests on one side of the head. The number of helices is calculated as the total number of all crests on both sides of the head divided by 2. Related to the Methods section.
(TIF)

**S1 Video. The motion of spermatozoa during the release from spermatozeugmata in three *Elasmobranchii* species.** Related to the results section "Sperm release from spermatozeugmata clusters" and Fig 3.
(MP4)

**S2 Video. Examples of cytoplasmic sleeve removal during the motion of spermatozoa.** Related to the results section "Cytoplasmic sleeve."
(MP4)

**S3 Video. Motility of *Elasmobranchii* spermatozoa in different viscosity media.** Related to the results sections "Sperm motility and progression are specific for the highly viscous environment," "Flagellar wave adaptation to viscosity, "and Fig 4.
(MP4)

**S4 Video. The motion of Elasmobranchii spermatozoa during direction changes and head steering.** Related to the results section "Steering long sperm chiral bodies in high viscosity" and Fig 8.
(MP4)

## Acknowledgments

The authors would like to thank the Fundación Oceanogràfic for collaborating to obtain some of the samples.

## Author contributions

**Conceptualization:** Sergii Boryshpolets, Borys Dzyuba, Hermes Bloomfield-Gadêlha, Juan F. Asturiano.

**Formal analysis:** Sergii Boryshpolets, Borys Dzyuba, Anatoliy Sotnikov.

**Investigation:** Sergii Boryshpolets, Borys Dzyuba, Pablo García-Salinas, Victor Gallego.

**Methodology:** Sergii Boryshpolets, Borys Dzyuba, Pablo García-Salinas, Hermes Bloomfield-Gadêlha, Victor Gallego.

**Resources:** Borys Dzyuba, Pablo García-Salinas, Juan F. Asturiano.

**Supervision:** Sergii Boryshpolets, Borys Dzyuba, Juan F. Asturiano.

**Visualization:** Sergii Boryshpolets, Pablo García-Salinas, Anatoliy Sotnikov.

**Writing – original draft:** Sergii Boryshpolets, Borys Dzyuba.

**Writing – review & editing:** Sergii Boryshpolets, Borys Dzyuba, Pablo García-Salinas, Hermes Bloomfield-Gadêlha, Victor Gallego, Anatoliy Sotnikov, Juan F. Asturiano.

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
