## [Decision Letter · Decision Letter 0]

3 Jan 2025

PONE-D-24-52456The ancient and helical architecture of Elasmobranchii's spermatozoa enables progressive motility in viscous environments.PLOS ONE

Dear Dr. Boryshpolets,

Thank you for submitting your manuscript to PLOS ONE. After careful consideration, we feel that it has merit but does not fully meet PLOS ONE’s publication criteria as it currently stands. Therefore, we invite you to submit a revised version of the manuscript that addresses the points raised during the review process.

We look forward to receiving your revised manuscript.

Kind regards,

Wilfried A. Kues, Ph.D.

Academic Editor

PLOS ONE

Journal Requirements:

3. Thank you for stating the following financial disclosure: Part of the work was carried out with the support of: VVI CENAKVA Research Infrastructure (ID 90099, MEYS CR, 2019–2023); "Biodiversity" (CZ.02.1.01./0.0/0.0/16_025/0007370 Reproductive and genetic procedures for preserving fish biodiversity and aquaculture), the Fundación Biodiversidad (PRCV00683), and the Project ELASMOREP PID2022-138847-I00 funded by MICIU/AEI/10.13039/501100011033 and ERDF/EU. PG-S has a contract from the European Union through the Operational Program of the European Social Fund (ESF) of the Comunitat Valenciana 2014–2020 ACIF 2018 (ACIF/2018/147). VG has a "Ramón y Cajal" contract (RYC2021-031558-I) funded by the Ministerio de Ciencia e Innovación (Spain) and the NextGenerationEU (European Union). 

4. Thank you for stating the following in the Acknowledgments Section of your manuscript: Part of the work was carried out with the support of: VVI CENAKVA Research Infrastructure (ID 90099, MEYS CR, 2019–2023); "Biodiversity" (CZ.02.1.01./0.0/0.0/16_025/0007370 Reproductive and genetic procedures for preserving fish biodiversity and aquaculture), the Fundación Biodiversidad (PRCV00683), and the Project ELASMOREP PID2022-138847-I00 funded by MICIU/AEI/10.13039/501100011033 and ERDF/EU. PG-S has a contract from the European Union through the Operational Program of the European Social Fund (ESF) of the Comunitat Valenciana 2014–2020 ACIF 2018 (ACIF/2018/147). VG has a "Ramón y Cajal" contract (RYC2021-031558-I) funded by the Ministerio de Ciencia e Innovación (Spain) and the NextGenerationEU (European Union). The authors would like to thank the Fundación Oceanogràfic for collaborating to obtain some of the samples. 

Please remove any funding-related text from the manuscript and let us know how you would like to update your Funding Statement. Currently, your Funding Statement reads as follows: Part of the work was carried out with the support of: VVI CENAKVA Research Infrastructure (ID 90099, MEYS CR, 2019–2023); "Biodiversity" (CZ.02.1.01./0.0/0.0/16_025/0007370 Reproductive and genetic procedures for preserving fish biodiversity and aquaculture), the Fundación Biodiversidad (PRCV00683), and the Project ELASMOREP PID2022-138847-I00 funded by MICIU/AEI/10.13039/501100011033 and ERDF/EU. PG-S has a contract from the European Union through the Operational Program of the European Social Fund (ESF) of the Comunitat Valenciana 2014–2020 ACIF 2018 (ACIF/2018/147). VG has a "Ramón y Cajal" contract (RYC2021-031558-I) funded by the Ministerio de Ciencia e Innovación (Spain) and the NextGenerationEU (European Union). 

Reviewers' comments:

Reviewer's Responses to Questions

**Comments to the Author**

1. Is the manuscript technically sound, and do the data support the conclusions?

Reviewer #1: Yes

2. Has the statistical analysis been performed appropriately and rigorously? 

Reviewer #1: N/A

3. Have the authors made all data underlying the findings in their manuscript fully available?

Reviewer #1: Yes

4. Is the manuscript presented in an intelligible fashion and written in standard English?

Reviewer #1: Yes

5. Review Comments to the Author

Reviewer #1: This is an excellent paper highlighting sperm motility in three cartilaginous fish species. The authors have described sperm motility and sperm structure both qualitatively as well as quantitatively. They have comprehensively shown the importance of investigating sperm motility at high viscosities, presumably mimicking parts of the female reproductive tract where fertilization is likely to occur. While I believe they have performed a novel study of great interest I think they have underplayed the osmotic effects at least of sea-water initiating further motility and hardly mention the really extremely important fact that sharks represent the only vertebrates having a blood osmotic concentration of about 1000mOsm/Kg or equivalent to sea-water. More attention should have been devoted both experimentally as well as in the discussion on the flushing of sperm using the siphon organ to flush sperm along the clasper groves into the female reproductive tract. The authors should further explain the already activated sperm that are further activated and pushed towards the more viscous female environment. So, the authors should at least expand their discussion to include this.

My other main criticism is that they have measured straight line length of sperm but not the true length. There are many simple image analysis programmes to accurately measure sperm head length following the helical sperm head length.

My other comments are mainly minor and relate to grammar and typography. While I will list some examples here they need to get someone who is in command of really good scientific English to edit the paper thoroughly for scientific English

Some examples will suffice

Line 39: low viscosity and real osmotic concentration

Line 40: Even in abstract write out abbreviations such as MC

Line 48: Modern not correct... should be currently or Surviving....

Line 132: not possible... is not a good reason

Line 173: Write abbreviations in full when explaining/discussing for the first time

Line 358: forces

Line 361: reference required

Lines 397 to 400: This refers to my arguments above and the balance between just sea-water activation, initial mixing with tubular fluid or experimental with methyl cellulose need to be better investigated/explained.

Line 448: At least speculate on these aspects and maybe it is still the attachment/remnants of sheaths related to Sertoli cell "embedding material"

Finally, I like the discussing/hypothesizing around the issue of sperm shape/motility/penetration through mucous and potentially fertilization outcome. The videos are excellent and help to follow the arguments,

6. PLOS authors have the option to publish the peer review history of their article (what does this mean? ). If published, this will include your full peer review and any attached files.

**Do you want your identity to be public for this peer review?** For information about this choice, including consent withdrawal, please see our Privacy Policy .

Reviewer #1: **Yes: ** Prof Gerhard van der Horst

---

## [Author Response · Author response to Decision Letter 0]

13 Jan 2025

Dear Editor and Reviewer,

We would like to express our sincere gratitude to the expert reviewers for their specialist time, for the thorough examination of our paper, and for the detailed feedback provided. We feel that the manuscript has greatly improved after the insightful and constructive comments by the referee.

Please find below our point-by-point responses to each individual comment (also provided in 'Response to Reviewers' file).

Reviewers'comments:

Reviewer's Responses to Questions

Comments to the Author

1. Is the manuscript technically sound, and do the data support the conclusions?

Reviewer #1: Yes

2. Has the statistical analysis been performed appropriately and rigorously?

Reviewer #1: N/A

3. Have the authors made all data underlying the findings in their manuscript fully available?

Reviewer #1: Yes

4. Is the manuscript presented in an intelligible fashion and written in standard English?

Reviewer #1: Yes

5. Review Comments to the Author

Reviewer #1: This is an excellent paper highlighting sperm motility in three cartilaginous fish species. The authors have described sperm motility and sperm structure both qualitatively as well as quantitatively. They have comprehensively shown the importance of investigating sperm motility at high viscosities, presumably mimicking parts of the female reproductive tract where fertilization is likely to occur. While I believe they have performed a novel study of great interest I think they have underplayed the osmotic effects at least of seawater initiating further motility and hardly mention the really extremely important fact that sharks represent the only vertebrates having a blood osmotic concentration of about 1000mOsm/Kg or equivalent to seawater. More attention should have been devoted both experimentally as well as in the discussion on the flushing of sperm using the siphon organ to flush sperm along the clasper groves into the female reproductive tract. The authors should further explain the already activated sperm that are further activated and pushed towards the more viscous female environment. So, the authors should at least expand their discussion to include this.

We sincerely thank Prof. Gerhard van der Horst for his thorough review of our manuscript and his valuable comments, as well as for his enthusiasm for this research. Below, we have provided detailed responses to each individual point.

We have now amended the text in the discussion section to reflect your suggestion, including more details: "All Elasmobranchii species are internal fertilizers. At the same time, the fertilization process itself is not fully understood in these species. Males deposit spermatozoa through specific grooves on claspers (extended pelvic fins) during mating [2]. However, these channels are not fully closed, allowing contact with the aquatic environment (seawater in marine species and freshwater in freshwater species). Also, the sperm is propelled into the female by the accumulation and subsequent release of water from the siphonal sacs in sharks [37, 38] or, presumably, by secretions from claspers glands in rays and skates [39]. Thus, sperm motility could potentially be activated by external factors similar to external fertilizers [40]. On the other hand, the unique physiology of marine Elasmobranchii species sets them apart from other vertebrates since their blood and seminal plasma osmolality is approximately 1000 mOsm/kg (equivalent to seawater), thus preventing osmotic shock during transition. In contrast, freshwater Elasmobranchii have internal body fluid osmolality ranging between 243–350 mOsm/kg. In this case, contact of spermatozoa with freshwater induces osmotic shock, potentially resembling signalling in freshwater external fertilizers. However, in our previous observations, freshwater led to a rapid cessation of flagellar beating [41]. These findings suggest that the motility of Elasmobranchii spermatozoa is not directly regulated by environmental osmolality. Nevertheless, further research is needed to explore the potential effects of the siphonal sac or clasper gland fluids on sperm physiology during mating and to uncover the precise mechanisms involved in sperm deposition and interaction with the female's internal environment. Our observations indicate that the spermatozoa flagellum remained actively beating in undiluted seminal fluid and ASF across all three species, even when spermatozoa were still compacted within the spermatozeugmata. When LVM was used, the spermatozoa were nonprogressive, tumbling in place, and struggled to leave the spermatozeugmata unless it was mechanically disrupted during mixing with ASF. Interestingly, an increase in ASF viscosity was sufficient to trigger progressive sperm motility. Thus, we suggested that during mating, the already activated spermatozoa (still packed in spermatozeugmata) are transported towards the female's internal organs, where the next phase of progressive motility is initiated upon encountering the more viscous internal environment."

My other main criticism is that they have measured straight line length of sperm but not the true length. There are many simple image analysis programmes to accurately measure sperm head length following the helical sperm head length.

Many thanks for this. Initially, we intended to measure the true length of sperm heads using image analysis software. However, we found that the 3D nature of the head and its irregular shape—varying helix pitch along the head—made these measurements biased and inaccurate. Thus, obtaining exact measurements that reflect the true length is challenging and would not influence the study's main findings. Instead, a straightforward approximation allows us to compare our data with previously published results, where straight-line length measurements were used. For example, see Tanaka S, Kurokawa H, Masako H. Comparative Morphology of the Sperm in Chondrichthyan Fishes. In: Jamieson B, G, Ausio J, Jean-Lou J, editors. Advances in Spermatozoal Phylogeny and Taxonomy. Muséum national d'Histoire naturelle, Paris: Editions du Muséum national d'Histoire naturelle; 1995. p. 313-20.

My other comments are mainly minor and relate to grammar and typography. While I will list some examples here they need to get someone who is in command of really good scientific English to edit the paper thoroughly for scientific English

Some examples will suffice

We are very grateful for pointing out this important omission in our manuscript. We have now endeavoured to remove typographical and grammar errors by thoroughly proofreading. Additionally, during manuscript preparation, the scientific English was thoroughly reviewed multiple times by us and our colleagues, and we believe it meets the standards of the journal. However, if the reviewer or editors feel it is necessary, we are open to submitting the manuscript for an additional professional English proofreading service.

Line 39: low viscosity and real osmotic concentration

The text of the abstract was corrected, and missing information was added

Line 40: Even in abstract write out abbreviations such as MC

The text was corrected, and abbreviations are described.

Line 48: Modern not correct... should be currently or Surviving....

The word "Modern" was replaced with "Currently living."

Line 132: not possible... is not a good reason

Replaced with "challenging"

Line 173: Write abbreviations in full when explaining/discussing for the first time

The text was corrected, and the abbreviations are now described.

Line 358: forces

The text was corrected.

Line 361: reference required

The reference was added: 27.Andrietti F, Bernardini G. The movement of spermatozoa with helical head: theoretical analysis and experimental results. Biophysical Journal. 1994;67(4):1767-74. doi: 10.1016/S0006-3495(94)80651-4.

Lines 397 to 400: This refers to my arguments above and the balance between just seawater activation, initial mixing with tubular fluid or experimental with methyl cellulose need to be better investigated/explained.

Many thanks for this suggestion. The text in the discussion section has now been extended to include more details per your suggestion.

Line 448: At least speculate on these aspects and maybe it is still the attachment/remnants of sheaths related to Sertoli cell "embedding material"

Thank you for this suggestion. The text is corrected and extended: "The Hermes body, or the cytoplasmic droplet, is a component of the sperm flagellum unique to epididymal spermatozoa, originating as the remnant of germ cell cytoplasm during spermatogenesis. Its precise role and function are unclear, but several authors hypothesize that this structure might be associated with the maturation process and acquisition of sperm motility [56, 57]. Alternatively, the observed sleeves could be reminiscent of the structural matrix that facilitates the interaction between Sertoli cells and developing spermatozoa during spermatogenesis since they are not a part of spermatozoa itself but rather just provide an additional layer or cover at the midpiece with this extra membrane."

Finally, I like the discussing/hypothesizing around the issue of sperm shape/motility/penetration through mucous and potentially fertilization outcome. The videos are excellent and help to follow the arguments,

We are incredibly thankful for your positive evaluation and enthusiastic feedback on our work.

6. PLOS authors have the option to publish the peer review history of their article (what does this mean?). If published, this will include your full peer review and any attached files.

Do you want your identity to be public for this peer review? For information about this choice, including consent withdrawal, please see our Privacy Policy.

Reviewer #1: Yes: Prof Gerhard van der Horst

---

## [Decision Letter · Decision Letter 1]

31 Jan 2025

The ancient and helical architecture of Elasmobranchii's spermatozoa enables progressive motility in viscous environments.

PONE-D-24-52456R1

Dear Dr. Boryshpolets,

We’re pleased to inform you that your manuscript has been judged scientifically suitable for publication and will be formally accepted for publication once it meets all outstanding technical requirements.

Kind regards,

Wilfried A. Kues, Ph.D.

Academic Editor

PLOS ONE

Additional Editor Comments (optional):

Reviewers' comments:

Reviewer's Responses to Questions

**Comments to the Author**

1. If the authors have adequately addressed your comments raised in a previous round of review and you feel that this manuscript is now acceptable for publication, you may indicate that here to bypass the “Comments to the Author” section, enter your conflict of interest statement in the “Confidential to Editor” section, and submit your "Accept" recommendation.

Reviewer #1: All comments have been addressed

2. Is the manuscript technically sound, and do the data support the conclusions?

Reviewer #1: Yes

3. Has the statistical analysis been performed appropriately and rigorously? 

Reviewer #1: Yes

4. Have the authors made all data underlying the findings in their manuscript fully available?

Reviewer #1: Yes

5. Is the manuscript presented in an intelligible fashion and written in standard English?

Reviewer #1: Yes

6. Review Comments to the Author

Reviewer #1: Well done and nice paper suggesting to be published now

7. PLOS authors have the option to publish the peer review history of their article (what does this mean? ). If published, this will include your full peer review and any attached files.

**Do you want your identity to be public for this peer review?** For information about this choice, including consent withdrawal, please see our Privacy Policy .

Reviewer #1: **Yes: ** Prof Gerhard van der Horst

---

## [Editor Report · Acceptance letter]

PONE-D-24-52456R1

PLOS ONE

Dear Dr. Boryshpolets,

I'm pleased to inform you that your manuscript has been deemed suitable for publication in PLOS ONE. Congratulations! Your manuscript is now being handed over to our production team.

Kind regards,

on behalf of

Dr. Wilfried A. Kues

Academic Editor

PLOS ONE